# Loss of histone methyltransferase ASH1L in the developing mouse brain causes autistic-like behaviors

Yuen Gao [1,5], Natalia Duque-Wilckens[2,5], Mohammad B. Aljazi[1], Yan Wu[1], Adam J. Moeser [2,3], George I. Mias [1,4], Alfred J. Robison[2] & Jin He [1✉]

Autism spectrum disorder (ASD) is a neurodevelopmental disease associated with various gene mutations. Recent genetic and clinical studies report that mutations of the epigenetic gene *ASH1L* are highly associated with human ASD and intellectual disability (ID). However, the causality and underlying molecular mechanisms linking *ASH1L* mutations to genesis of ASD/ID remain undetermined. Here we show loss of ASH1L in the developing mouse brain is sufficient to cause multiple developmental defects, core autistic-like behaviors, and impaired cognitive memory. Gene expression analyses uncover critical roles of ASH1L in regulating gene expression during neural cell development. Thus, our study establishes an ASD/ID mouse model revealing the critical function of an epigenetic factor ASH1L in normal brain development, a causality between *Ash1L* mutations and ASD/ID-like behaviors in mice, and potential molecular mechanisms linking *Ash1L* mutations to brain functional abnormalities.

[1] Department of Biochemistry and Molecular Biology, College of Natural Science, Michigan State University, East Lansing, MI, USA. [2] Department of Physiology, College of Natural Science, Michigan State University, East Lansing, MI, USA. [3] Gastrointestinal Stress Biology Laboratory, Department of Large Animal Clinical Sciences, College of Veterinary Medicine, East Lansing, MI, USA. [4] Institute for Quantitative Health Science and Engineering, Michigan State University, East Lansing, MI, USA. [5] These authors contributed equally: Yuen Gao, Natalia Duque-Wilckens. ✉email: hejin1@msu.edu

Autism spectrum disorder (ASD) is one of the most prevalent neurodevelopmental disorders (NDDs) that have a strong genetic basis[1]. As hundreds of new ASD risk genes have been identified by recent genetic studies[2–6], developing gene-specific knockout animal models has emerged as a priority for further determining whether the identified ASD risk genes are the causative drivers leading to ASD development, as well as for understanding the biological mechanisms underlying the pathogenesis caused by the mutations of ASD risk genes.

ASH1L (Absent, Small, or Homeotic 1-Like) protein is a histone methyltransferase that mediates di-methylation of histone H3 lysine 36[7]. Similar to the function of other members of Trithorax-group (TrxG) proteins, ASH1L facilitates gene expression during normal development[8]. Recent genetic studies on large cohorts of ASD patients reported that mutations of ASH1L are highly associated with human ASD[2–4,6]. The genetic findings are supported by multiple clinical reports that some children diagnosed with ASD and/or intellectual disability (ID) acquire various disruptive or missense mutations of ASH1L[9–14]. In addition to ASD and ID, patients also display a variety of developmental and behavioral abnormalities including delayed myelination, microcephaly, craniofacial deformity, skeletal abnormality, and feeding difficulties, suggesting critical roles of ASH1L in normal embryonic and postnatal development[9,11,12]. However, two fundamental questions regarding ASH1L mutations and genesis of ASD/ID remain to be elucidated: (i) is loss of ASH1L in the developing brain sufficient to induce ASD/ID-related phenotypes; and (ii) what are the molecular mechanisms linking ASH1L mutations to the pathogenesis of ASD/ID?

In this study, we used an Ash1L knockout mouse model to show that deletion of ASH1L in the developing mouse brain is sufficient to cause ASD/ID-like behaviors, confirming disruptive ASH1L mutations are the causative drivers for the development of ASD/ID. At the molecular level, loss of ASH1L impairs the expression of genes critical for normal brain development, indicating mis-regulation of ASH1L-mediated gene expression during brain development is likely to be a key molecular mechanism linking ASH1L mutations to ASD/ID development.

## Results

**Generation and characterization of Ash1L knockout mice.** To examine the function of Ash1L in mouse development, we generated an Ash1L conditional knockout (cKO) mouse line by inserting two LoxP elements into the exon 4-flanking sites at the Ash1L gene locus (Ash1L[+/2f]). A CRE recombinase-mediated deletion of exon 4 resulted in altered splicing of mRNA that created a premature stop codon before the sequences encoding the first functional AWS (Associated With SET) domain. The truncated ASH1L protein contained the N-terminal 1,694 amino acids but lost all functional domains, thus mimicking the disruptive mutations found in patients (Fig. 1a and Supplementary Fig. 1a, b). To preclude mouse strain-specific effects on animal phenotypic and behavioral changes, we backcrossed the wild-type Ash1L[+/2f] founders with C57BL/6 mice for more than five generations to reach a pure genetic background. The heterozygous Ash1L-KO mice (Ash1L[+/1f]) were obtained by crossing the wild-type Ash1L[+/2f] mice with CMV-Cre mice, through which one allele of Ash1L gene was deleted in both germlines and somatic cells in progenies. The heterozygous Ash1L[+/1f] x heterozygous Ash1L[+/1f] mating produced normal numbers of embryos. The gross embryos and placentas did not show obvious differences between wild-type and global Ash1L-KO (Ash1L[1f/1f]) embryos at embryonic day 13.5 (E13.5), and all embryos developed to term with expected Mendelian ratios (Supplementary Fig. 1c, d), suggesting Ash1L was dispensable for mouse embryonic

development. The global Ash1L-KO newborns displayed similar body size and weight to their wild-type littermates at postnatal day 0 (P0) (Supplementary Fig. 1e). However, without maternal uterine support, all Ash1L-KO newborns died within 24 h after birth (Fig. 1b and Supplementary Fig. 1d), suggesting Ash1L might be critical for establishing and maintaining a stable physiological condition for neonatal survival. Further anatomical analyses did not reveal obvious gross morphological abnormalities of individual organs, except that the majority of Ash1L-KO newborns displayed aberrant rib numbers (Fig. 1c), which was consistent with the function of TrxG proteins in body segmentation and skeletal formation during embryonic development[15].

To examine the function of Ash1L in the development of central nervous system, we deleted Ash1L in the developing mouse brain by crossing the Ash1L-cKO mice with a neural progenitor cell (NPC)-specific Cre (Nestin-Cre) mouse line. The CRE recombinase expressed in the NESTIN[+] cells induced Ash1L deletion in NPCs as well as NPC-derived neuronal and glial lineages in the embryonic developing mouse brain. The mating (Ash1L[2f/2f];Nestin-Cre[−/−] x Ash1L[2f/+];Nestin-Cre[+/−]) produced wild-type (Ash1L[2f/2f];Nestin-Cre[−/−]), heterozygous (Ash1L[2f/+];Nestin-Cre[+/−]), and homozygous Ash1L-cKO (Ash1L-Nes-cKO, Ash1L[2f/2f];Nestin-Cre[+/−]) progenies with expected Mendelian ratios. Compared to the wild-type littermates, the homozygous Ash1L-Nes-cKO newborns had similar body weight at birth and survived through early postnatal days. However, the homozygous Ash1L-Nes-cKO pups gradually displayed growth retardation, indicated by both smaller body size and lower body weight. The observed postnatal growth retardation appeared to be more drastic 2–3 weeks after birth, and the average body weight of Ash1L-Nes-cKO pups was approximately 50% less than that of wild-type littermates at P21 (Fig. 1d, e). Although around 10% Ash1L-Nes-cKO pups died before weaning, the majority of surviving pups were able to grow to adulthood with their final body weight 5–10% lower than that of their wild-type adult littermates (Fig. 1f). Similar to the craniofacial deformity observed in patients[4,9,11,12], the adult Ash1L-Nes-cKO mice displayed an abnormal craniofacial appearance with a reduced eye-to-mouth distance, which was caused by shortened nose bones revealed by both micro-CT scan and skull bone staining (Fig. 1g–j). No other obvious gross abnormalities of individual organs were observed in the Ash1L-Nes-cKO adult mice.

**Loss of ASH1L delays embryonic and postnatal brain development.** Next, we set out to investigate microscopic changes of Ash1L-KO mouse brains. Although the global Ash1L-KO newborns (P0) displayed normal gross brain appearance and size (Supplementary Fig. 2a, b), Nissl staining showed that the cortices of global Ash1L-KO mice were disorganized and lost its minicolumnar arrangement (Fig. 2a), indicating malformations of cortical development (MCD). To further examine whether the disorganized cortices were caused by aberrant lamination during embryonic cortical development, we analyzed each cortical layer formation by immunostaining with cortical layer-specific antibodies. The results showed that the majority of layer II-III (L2/3)-specific SATB2[+] neurons were located on the L2/3 in the wild-type cortices at P0. In contrast, some SATB2[+] neurons in the Ash1L-KO cortices were not properly located on the upper layers and scattered in the bottom layers (Fig. 2b, c), suggesting that Ash1L deletion in the developing mouse brain resulted in the delayed lamination of neuronal cells during embryonic cortical layer formation.

To examine whether the Ash1L deletion in developing brains could lead to delayed myelination in postnatal mouse brains as observed in some ASH1L-mutation-related ASD/ID patients[11],

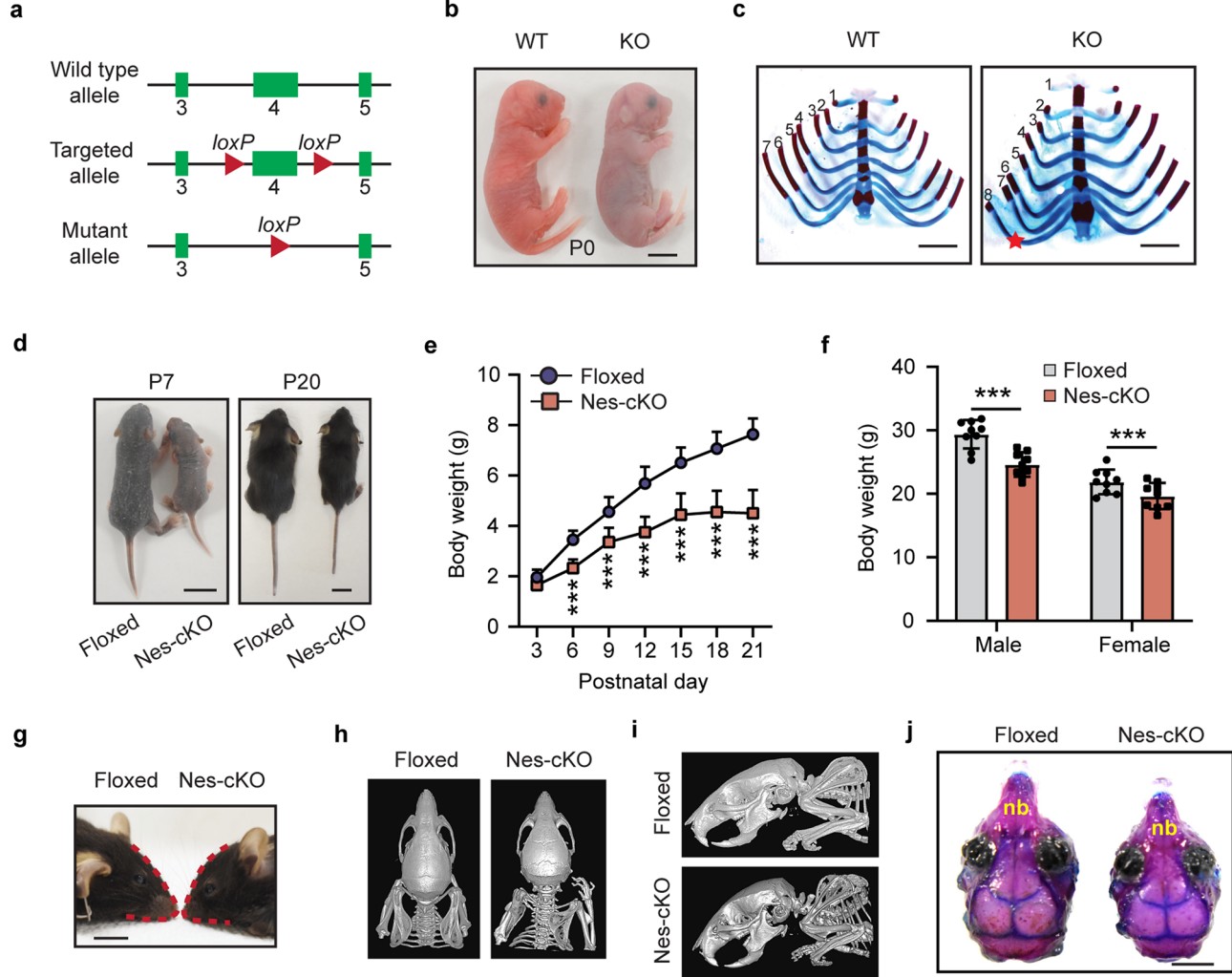

**Fig. 1 Characterization of *Ash1L* knockout mice. a** Diagram showing the strategy for the generation of *Ash1L* conditional knockout mice. **b** Representative photos of wild-type and global *Ash1L*-KO newborns at P0. The *Ash1L*-KO newborns died at P0, bar = 5 mm. **c** Photos showing the ventral view of rib cages of wild-type and *Ash1L*-KO mice, bar = 2 mm. **d** Representative photos showing the body size of wild-type and *Ash1L*-Nes-cKO mice at P7 and P20, bar = 1 cm. **e** Postnatal growth curve of wild-type and *Ash1L*-Nes-KO mice before weaning. Mixed gender body weight was plotted. For each group, $n = 15$ mice. *P*-values calculated using a two-way ANOVA test. Error bars in graphs represent mean ± SEM. Note: **$p < 0.01$; ***$p < 0.001$. **f** Body weight of adult wild-type and *Ash1L*-Nes-cKO mice. All the mice were measured at 3-month old. For each group, $n = 9$ mice. *P*-values calculated using a two-way ANOVA test. Error bars in graphs represent mean ± SEM. Note: ***$p < 0.001$. **g** Photo showing the craniofacial deformity of *Ash1L*-Nes-cKO mice, bar = 5 mm. **h–i** A dorsal (h) and lateral (i) view of mouse skull shown by micro-CT scanning. **j** A dorsal view of skull shown by bone staining. nb, nose bone, bar = 2 mm.

we performed immunostaining for myelin basic protein (MBP) to examine the dynamic myelination in the postnatal developing brain. The results showed that both wild-type and homozygous *Ash1L*-Nes-cKO cortices lacked discernable myelination at P0 (Fig. 2d). However, compared to the wild-type controls that had increased myelination over time, the levels of myelination in the *Ash1L*-Nes-cKO cortices were significantly lower at P21 but reached to a comparable level around postnatal two months (Fig. 2d, e), suggesting the *Ash1L* deletion in the developing brain led to delayed myelination during early postnatal brain development.

**Loss of ASH1L in the developing mouse brain causes ASD/ID-like behaviors.** After characterizing the gross phenotypes of global and conditional *Ash1L* KO mice, we set out to investigate whether the *Ash1L* deletion in the developing brain could lead to abnormal behaviors in adult mice. Since deficits in social interaction as well as repetitive and restricted behaviors are two major

clinical manifestations found in human ASD patients, we first focused on testing these core autism-like behaviors. To this end, we performed a three-chamber test to examine the voluntary exploration of a social vs. a non-social stimulus (sociability) and the voluntary exploration of a familiar vs. a novel social stimulus (social novelty)[16,17]. In the sociability portion of the test (Fig. 3a), the results showed the main effects of stimulus type (F1,70 = 12.28, $p < 0.001$) and genotype (F1,70 = 6.34, $p = 0.01$). Planned comparisons revealed that the wild-type controls spent more time with the social stimulus than with the object ($p = 0.001$), representing normal sociability. In contrast, the homozygous *Ash1L*-Nes-cKO mice did not show a preference for the social stimulus, suggesting impaired sociability (Fig. 3b). In the social novelty portion of the test (Fig. 3c), there was an effect of genotype by stimulus interaction (F1,68 = 7.929, $p = 0.006$). Planned comparisons showed that the wild-type mice preferred a novel over a familiar animal ($p = 0.0001$), indicating a preference for social novelty. In contrast, the *Ash1L*-Nes-cKO mice showed no preference, indicating reduced social memory or lost interest in

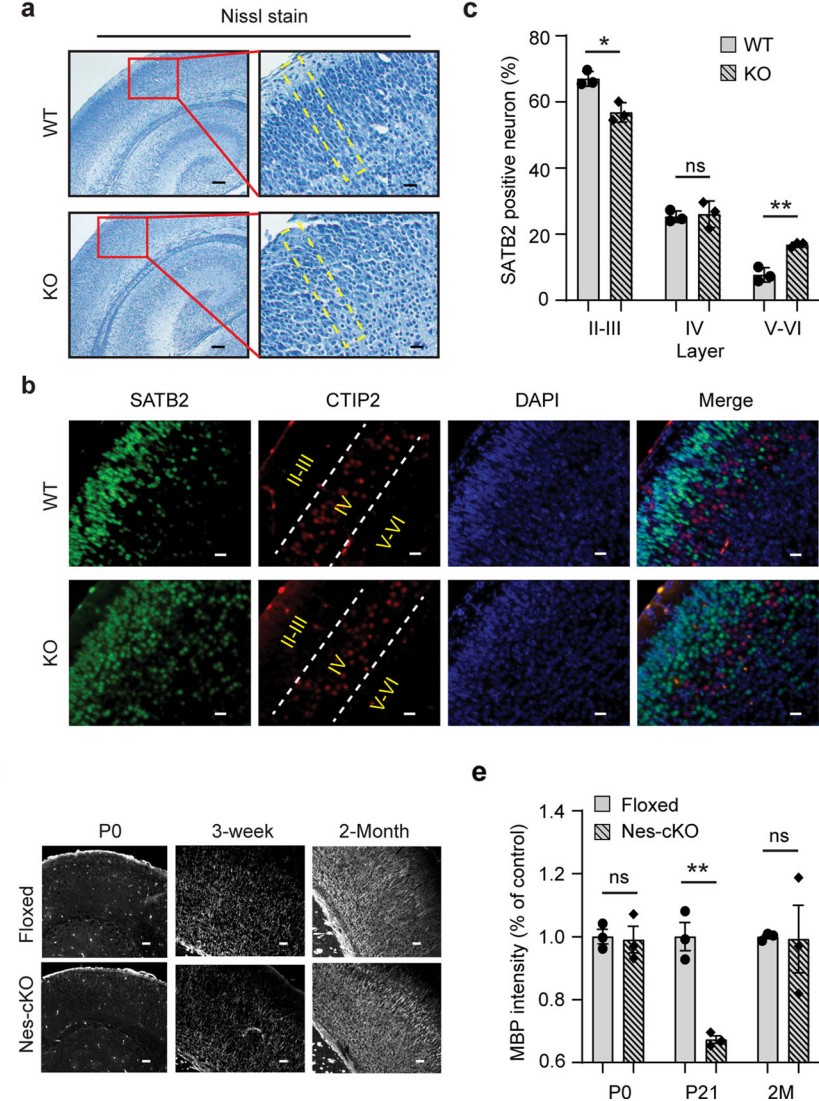

**Fig. 2 Loss of ASH1L delays embryonic and postnatal brain development. a** Nissl staining showing the cortical histology of wild-type and global *Ash1L*-KO newborns. The mini-columnar arrangement of cortical cells is highlighted by yellow lines. Bar (left) = 100 μm, Bar (right) = 20 μm. **b** Photos showing the distribution of L2/3-specific SATB2+ cells and L4-specific CTIP2+ cells in wild-type and *Ash1L*-KO cortices at P0, bar = 20 μm. **c** Quantification of SATB2+ neurons in different layers. For each group, *n* = 3 biologically independent samples. *P*-values calculated using a two-tailed *t* test. Error bars in graphs represent mean ± SEM. Note: \**p* < 0.05; \*\**p* < 0.01; ns, not significant. **d** Photos showing the myelin basic protein (MBP) staining of P0, P21, and postnatal 2-month cortices, bar = 100 μm. **e** Quantitative MBP expression analyzed by the integrated fluorescence intensity in cortices. Values are percentage of control values ± SEM. For each group, *n* = 3 biologically independent samples. *P*-values calculated using a two-tailed *t* test. Note: \*\**p* < 0.01; ns, not significant.

social novelty (Fig. 3d). In addition to impaired social interaction, we observed that all *Ash1L*-Nes-cKO adult mice displayed mild or severe hind paw clasping (*t* = 10.62, df = 28, *p* = 2.5e−11) when suspended by tails (Fig. 3e–g and Supplementary Movies 1–3), as well as increased overall grooming episodes (*t* = 4.66, df = 18, *p* = 0.0012) and time (*t* = 4.20, df = 18, *p* = 0.0023) that resulted in skin lesions (Fig. 3h–j and Supplementary Movies 4, 5), suggesting the *Ash1L* deletion caused repetitive and compulsive behaviors, one of the core clinical manifestations observed in human ASD patients.

Besides two core autistic behaviors, some patients with *ASH1L* mutations had ID as their main clinical manifestation[10–13]. Therefore, we further used the novel object recognition (NOR) test to examine whether the *Ash1L* gene deletion in the developing mouse brain could lead to impaired object memory (Fig. 3k). 24 h after initial familiarization with two identical

objects in an arena, the mice were allowed to explore the same arena in the presence of a familiar object and a novel object (Fig. 3k). The results showed that, compared to wild-type mice, the homozygous *Ash1L*-Nes-cKO mice showed a reduced discrimination index (*t* = 2.420, df = 33, *p* = 0.02) (Fig. 3l), indicating impaired memory. Furthermore, because some ASD/ID patients have elevated levels of anxiety, we examined anxiety-like behaviors by measuring time in the center of the arena during the first 10 min of habituation (Fig. 3k). Compared to wild-type controls, *Ash1L*-Nes-cKO spent about 50% less time exploring the arena center (*t* = 2.914, df = 36, *p* = 0.006) (Fig. 3m), indicating the *Ash1L*-Nes-cKO mice had increased anxiety-like behaviors. During the 5-minute period, *Ash1L*-Nes-cKO showed no difference in locomotor activity (Fig. 3n), indicating that the differences observed in social behaviors and object memory were not caused by altered locomotor activity. However, when we

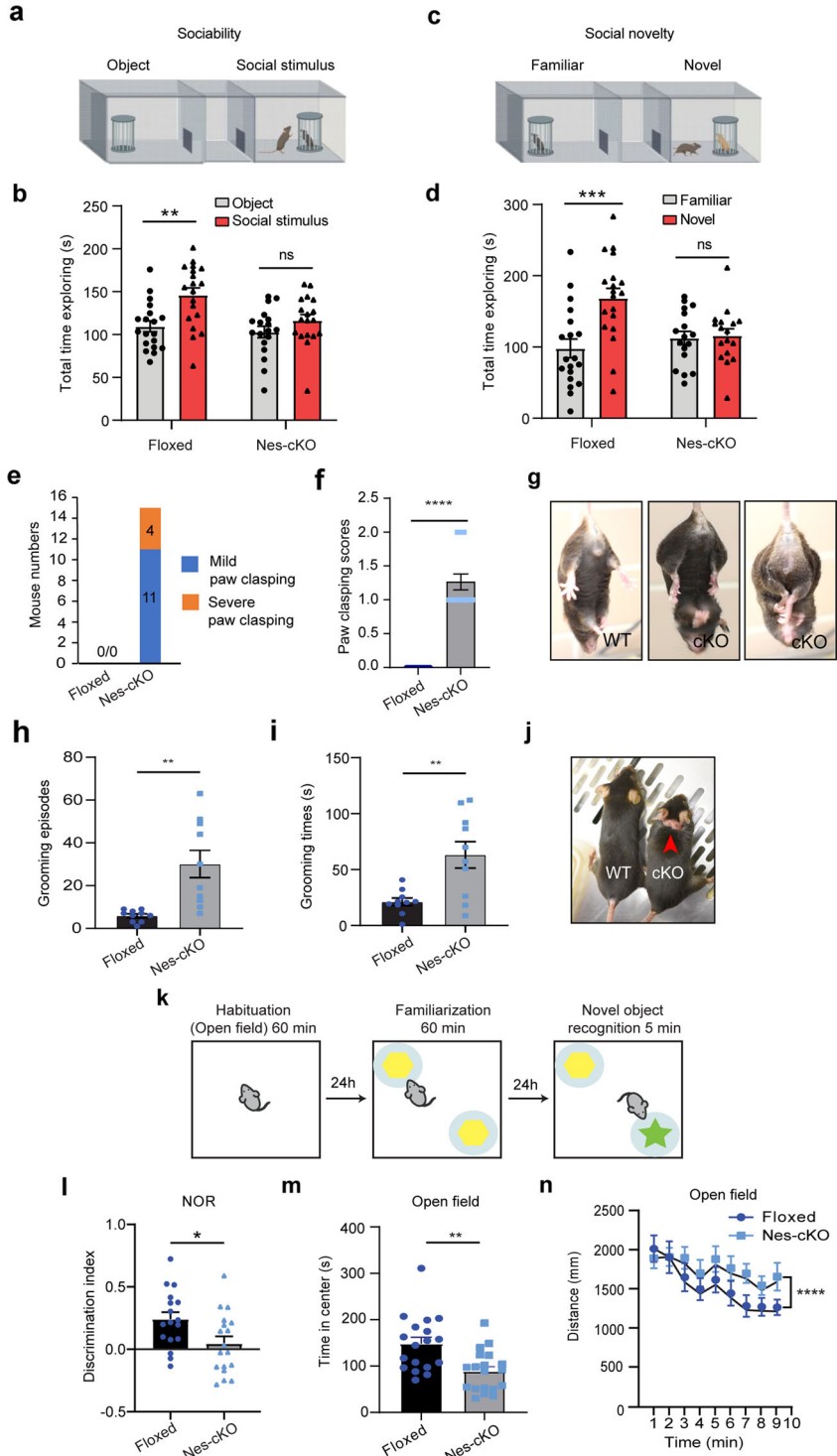

extended observation time to 10 min, the *Ash1L*-Nes-cKO mice show increased locomotor activity compared to wild type ($t =$ 2.496, df $=$ 36, $p = 0.017$) (Fig. 3n), suggesting that they may be hyperactive, a phenotype also sometimes found in autism patients.

Thus, the collective observations and behavior tests revealed that the loss of *Ash1L* in the developing mouse brain resulted in both autistic-like behaviors and ID-like defects, which were featured by reduced sociability, loss of interest in social novelty, repetitive and compulsive behaviors, impaired recognition memory, and increased anxiety-like behaviors.

**Loss of ASH1L impairs expression of genes critical for brain development**. ASH1L is an epigenetic factor important for activating gene expression during development[8]. The mouse ENCODE transcriptome data showed that *Ash1L* is highly expressed in the embryonic and adult mouse brain, suggesting its possible role in brain development and function[18]. To identify the genes regulated by ASH1L in the mouse brain development, we performed RNA-sequencing (RNA-seq) analyses to examine differential gene expression between wild-type and *Ash1L*-deleted neural cells. To reduce the complexity in gene expression analyses on bulk brains containing heterogenous neural lineages or the

**Fig. 3 Loss of ASH1L in the developing mouse brain causes ASD/ID-like behaviors. a** 3-chamber tests for sociability. **b** Quantitative results showing the time wild-type (floxed) and *Ash1L*-Nes-cKO mice spent in the chamber containing a social partner in the 3-chamber sociability tests. **c** 3-chamer tests for social novelty. **d** Quantitative results showing the time wild-type (floxed) and *Ash1L*-Nes-cKO mice spent in the chamber containing a novel animal in the 3-chamber social novelty tests. For the 3-chamber sociability and social novelty tests (**a–d**), *Ash1L* wild-type mice, $n = 19$; *Ash1L*-Nes-cKO mice, $n = 17$. *P*-values calculated using two-way ANOVA test. Error bars in graphs represent mean ± SEM. Note: **$p < 0.01$; ***$p < 0.001$; ns, not significant. **e** The numbers of wild-type and *Ash1L*-Nes-cKO mice showing mild or severe paw clasping. $n = 15$ mice/genotype. **f** Paw clasping scores of wild-type and *Ash1L*-Nes-cKO mice. $n = 15$ mice/genotype. Error bars in graphs represent mean ± SEM. *P*-values calculated using a two-tailed *t* test Note: ****$p < 0.0001$. **g** Compared to the wild-type mice (left panel), the *Ash1L*-Nes-cKO mice display mild (middle panel) or severe (right panel) paw clasping when suspended by tails. **h** Total grooming episodes in 10 min. **i** Total grooming time in 10 min. For the quantitative grooming measurement (**h–i**), $n = 10$ mice/genotype. *P*-values calculated using a two-tailed *t* test. Error bars in graphs represent mean ± SEM. Note: **$p < 0.01$. **j** Compared to the wild-type mice, the *Ash1L*-Nes-cKO mice display skin lesions caused by over-grooming (red arrow). **k** Open field and novel object recognition (NOR) tests. **l** The quantitative discrimination ratio of NOR tests. The discrimination ratio was calculated as (time spent on the novel object-time spent on the familiar object)/total time. *Ash1L* wild-type mice, $n = 17$; *Ash1L*-Nes-cKO mice, $n = 18$. *P*-values calculated using a two-tailed *t* test. Error bars in graphs represent mean ± SEM. Note: *$p < 0.05$. **m** Time spent in the center of the open field arena measured in 10-min habituation. **n** Total distance traveled per min in 10-min habituation. For the open field tests (**m–n**), *Ash1L* wild-type mice, $n = 18$; *Ash1L*-Nes-cKO mice, $n = 19$. *P*-values calculated using a two-tailed *t* test. Error bars in graphs represent mean ± SEM. Note: **$p < 0.01$, ****$p < 0.0001$.

same cell lineages located at different developmental stages, we set out to generate a tamoxifen-inducible *Ash1L*-cKO mouse line (*Ash1L^{2f/2f}*;*Rosa26-CreER^{T2+/+}*) by crossing the *Ash1L*-cKO mice with a Rosa26-CreER^{T2} line. The NPCs were isolated from the subventricular zone (SVZ) of brains and maintained in serum-free NPC culture medium. The deletion of *Ash1L* gene in the established NPCs was induced by 4-hydroxytamoxifen (4OH-TAM) added in the medium for 10 days (Supplementary Fig. 3a). Quantitative reverse transcription PCR (qRT-PCR) and western blot (WB) analyses showed *Ash1L*/ASH1L expression reduced to less than 5% at mRNA and protein levels in the *Ash1L*-KO NPCs (Supplementary Figs. 3b, c, 5 and 6). Both wild-type and *Ash1L*-KO NPCs were further induced to differentiate to neuronal lineages according to a well-established protocol[19], and immunostaining with lineage-specific markers was used to monitor the differentiation process. The results showed that both wild-type and *Ash1L*-KO cells expressed an NPC-specific marker NESTIN but not differentiation markers TUJ1 (neuron-specific tubulin III) or GFAP (glial fibrillary acidic protein) at day 0, indicating a homogenous NPC population before induced differentiation. 4 days after induced differentiation, both wild-type and *Ash1L*-KO cells had comparable decreased NESTIN^+ NPCs and increased TUJ1^+/GFAP^+ differentiated neural cells (Supplementary Fig. 3d–g).

To identify the differentially expressed genes in early NPC differentiation, we performed RNA-seq analyses 0, 12, and 24 h after induced differentiation. The results identified 2475 upregulated and 2808 downregulated genes during induced differentiation of wild-type NPCs (cutoff: fold changes > 1.5, $p < 0.01$) (Supplementary Fig. 4a, b). Gene ontology (GO) enrichment analyses showed the upregulated genes had enriched GO terms involving nervous system development, while the downregulated genes involved metabolic processes and cell cycle regulation (cutoff: FDR < 0.05) (Supplementary Fig. 4c, d and Supplementary Data 1, 2), reflecting the dynamic neural lineage development, cell cycle exit, and reduced metabolism during NPC differentiation. In the group of 2808 genes downregulated in the differentiating wild-type NPCs, 44 genes were found to have increased expression in the *Ash1L*-KO cells (cutoff: fold changes > 1.5, $p < 0.01$) (Fig. 4a), and the GO enrichment analysis showed that they were involved in cell migration, motility, and developmental growth (cutoff: FDR < 0.05) (Fig. 4b and Supplementary Table 1). In contrast, among the 2475 genes upregulated in the differentiating wild-type NPCs, 70 genes were found to have significantly reduced expression in the *Ash1L*-KO cells (cutoff: fold changes > 1.5, $p < 0.01$) (Fig. 4c), which had enriched GO terms involving telencephalon development, regulation of cell

communication, brain development, and central nervous development (cutoff: FDR < 0.05) (Fig. 4d and Supplementary Table 2). Consistent with the results of GO enrichment analyses, multiple genes downregulated in the *Ash1L*-KO cells, such as *Emx2*, *Dbx2*, *Pcdh10*, *Sall3*, and *Foxg1*, were previously reported to be involved in normal brain development and NDDs (Fig. 4e)[20–27]. To further validate the results in vivo, we performed the qRT-PCR analysis to examine the expression of these five NDD-related genes in wild-type and *Ash1L*-KO E16.5 cortices. The results showed that *Emx2*, *Pcdh10*, and *Foxg1* had similar reduced expression in both *Ash1L*-KO cortices and differentiating NPCs (Fig. 4f–i), while *Dbx2* and *Sall3* did not show significant difference in expression between wild-type and *Ash1L*-KO cortices (Supplementary Fig. 4e, f).

**Discussion**

In this study, we used an animal model to demonstrate that the loss of *Ash1L* gene alone in the developing mouse brain is sufficient to cause autistic-like behaviors and ID-like deficits in adult mice (Fig. 3), which strongly suggests disruptive *ASH1L* gene mutations found in patients are likely to be the causative drivers leading to clinical ASD/ID[9–14]. In addition, the early postnatal lethality found in the global *Ash1L*-KO newborns and the postnatal growth retardation observed in the *Ash1L*-Nes-cKO pups (Fig. 1d–f and Supplementary Fig. 1d) suggest that *Ash1L* might also play important roles in establishing neural circuits in the developing hypothalamus, which is critical for normal feeding behaviors and early postnatal growth[28]. It will be interesting to examine whether *ASH1L* mutations could cause hypothalamus dysfunctions that affect normal feeding behaviors and postnatal growth in human patients. Like the craniofacial deformity observed in the ASD/ID patients with *ASH1L* mutations, the *Ash1L*-Nes-cKO mice display craniofacial deformity with shortened nose bones (Fig. 1g–j). Since, in our animal model, *Ash1L* is also deleted in the NESTIN^+ neural crest stem cells (NCSCs) that develop into craniofacial skeletal tissues[29], it is possible that the observed craniofacial deformity is caused by aberrant craniofacial skeletal formation from the *Ash1L*-deleted NCSCs. At the microscopic level, we observed both delayed lamination of cortical neurons and myelination formation during embryonic and early postnatal brain development (Fig. 2), suggesting the loss of *Ash1L* in NPCs leads to delayed brain development in both neuronal and glial lineages.

Biochemically, ASH1L is a histone H3 lysine 36-specific methyltransferase that facilitates gene expression through regulating transcriptional activation[7]. To examine the molecular

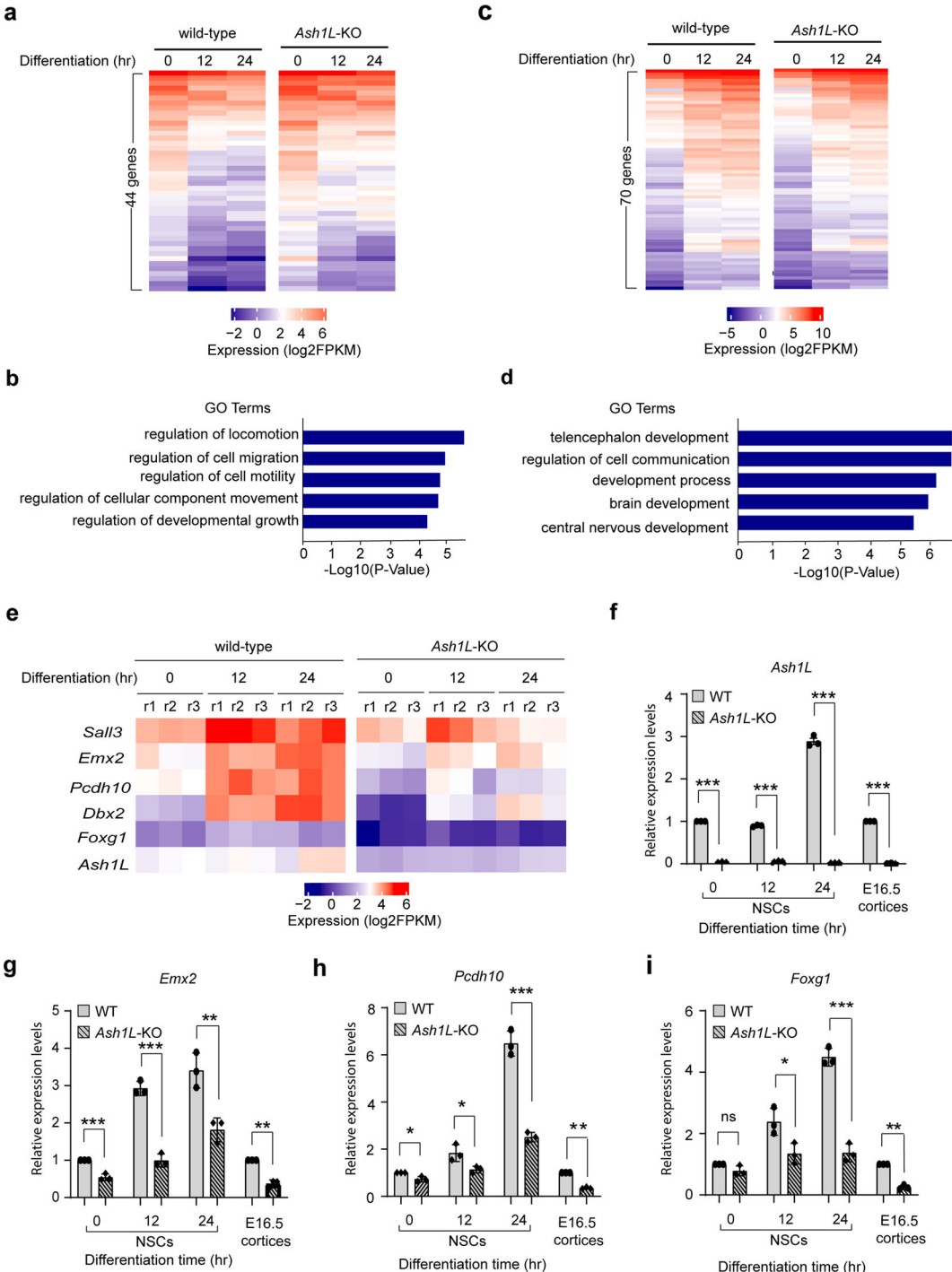

**Fig. 4 Loss of ASH1L impairs expression of genes critical for brain development. a** Heatmap showing 44 genes downregulated in the differentiating wild-type NPCs have significant increased expression in the *Ash1L*-KO cells. **b** Gene ontology enrichment analysis showing the enriched GO terms of 44 genes upregulated in the *Ash1L*-KO cells during differentiation (FDR < 0.05). **c** Heatmap showing 70 genes upregulated in the differentiating wild-type NPCs have significant reduced expression in the *Ash1L*-KO cells. **d** Gene ontology enrichment analysis showing the enriched GO terms of 70 genes downregulated in the *Ash1L*-KO cells during differentiation (FDR < 0.05). **e** Heatmap showing the representative NDD-related genes downregulated in the *Ash1L*-KO cells during differentiation (cutoff: fold change > 1.5, *p* < 0.01). **f–i** qRT-PCR analysis showing the mRNA levels of *Ash1L*, *Emx2*, *Pcdh10*, and *Foxg1* in wild-type and *Ash1L*-KO NPCs at different time points of induced differentiation and in bulk E16.5 cortices. The results of analysis in NPCs were normalized against levels of *Gapdh* and the expression level of wild-type NPCs at differentiation time 0 was arbitrarily set to 1. The results of analysis in E16.5 cortices were normalized against levels of *Gapdh* and the expression level of wild-type cortices was arbitrarily set to 1. *P*-values calculated using a two-tailed *t* test. The error bars represent mean ± SEM, *n* = 3 biologically independent samples/genotype. Note: *\*p* < 0.05; \*\**p* < 0.01; \*\*\**p* < 0.001; ns, not significant.

mechanisms underlying the *Ash1L*-deletion-induced brain developmental and functional abnormalities, we performed the gene expression analysis and identified 44 upregulated and 70 downregulated genes in the *Ash1L*-KO differentiating NPCs (Fig. 4a, c). Further analyses of gene functions did not reveal any significant NDD-related genes in the 44 upregulated genes (Fig. 4b and Supplementary Table 1). In contrast, multiple genes critical for normal brain development and highly related to human ASD/ID were found to have reduced expression in the differentiating *Ash1L*-KO NPCs (Fig. 4d, e and Supplementary Table 2), suggesting that impaired expression of neurodevelopmental genes is likely to be a main molecular mechanism linking *ASH1L* mutations to abnormal brain development. Of note, loss of *Emx2* was reported to impair the neuronal migration in the developing mouse cortex[30], which might cause the delayed cortical lamination observed in the *Ash1L*-KO developing cortex (Fig. 2b, c). In addition, mutations of *FOXG1* gene, one of the genes downregulated in the *Ash1L*-KO cells, result in human FOXG1 syndrome. Interestingly, FOXG1 syndrome and *ASH1L* mutations-induced ASD/ID have highly overlapping clinical manifestations[26], suggesting that ASH1L might function as a master epigenetic regulator to facilitate the expression of *FOXG1* and other critical genes for normal brain development, while mutations of *ASH1L* lead to mis-regulation of gene expression, disturbance of the normal brain developmental program, and brain functional abnormalities of ASD/ID.

Our current study has following limitations: (i) since the heterozygous *Ash1L*-Nes-cKO mice are not included in this study, it is unclear whether the degree of ASD/ID-like defects is correlated with *Ash1L* gene dosage in the developing brain; (ii) since bulk cortical tissues are used to validate the differential gene expression identified in the cultured differentiating NPCs, the discrepancy of results (Supplementary Fig. 4e, f) could be caused by heterogenous cell lineages or different developmental status of neural cells in the bulk cortical tissues. In future studies, the heterozygous *Ash1L*-Nes-cKO mice will be included to examine whether loss of a single *Ash1L* copy in the developing brain is sufficient to cause ASD/ID-like phenotypes. Furthermore, dynamic gene expression analysis of the neural lineage-specific cells directly isolated from developing brains will provide mechanistic insight into the function of ASH1L in brain development and pathogenesis of ASD/ID in vivo.

Finally, the ASD/ID mouse model generated in this study recapitulates most clinical ASD/ID manifestations found in human patients, which provides an invaluable tool for further exploring the biological mechanisms underlying the pathogenesis of *ASH1L*-mutation-induced ASD/ID, developing and testing new therapeutic approaches based on the function of ASH1L in regulating brain developmental gene expression revealed by this study.

## Methods

**Mice.** The *Ash1L* conditional knockout target construct was generated by modifying the BAC clone (RP24-394C15) based on the Recombineering method[31]. Two LoxP elements were inserted into the exon 4-flanking sites. The targeting construct was electroporated into the C57BL/6:129 hybrid murine ES cells. Homologous recombinant ES cell clones were identified PCR-based genotyping and injected into blastocysts. The genetically modified ES cells were micro-injected to blastocysts and transferred to the uterus of CD-1 pseudo-pregnant females to generate chimeric founder mice. The chimeric founder mice were crossed to a FLP recombinase mouse line to remove the FRT-flanked selection cassette. All mice were backcrossed to C57BL/6 mice for at least five generations to reach a pure C57BL/6 background before further mating to specific Cre lineages. Mice were housed under standard conditions (12 h light: 12 h dark cycles) with food and water ad libitum. The data obtained from all embryos were pooled without discrimination of sexes for the analysis. All mouse experiments were performed with the approval of the Michigan State University Institutional Animal Care & Use Committee.

**Mouse breeding strategy**. All mice were backcrossed to C57BL/6 mice for at least five generations to reach a pure C57BL/6 background. (1) Generating *Ash1l* global knockout mice: The heterozygous *Ash1L*-KO mice ($Ash1L^{+/1f}$) were obtained by crossing the wild-type $Ash1L^{+/2f}$ mice with CMV-Cre mice (B6.C-Tg (CMV-cre) 1Cgn/J, The Jackson Laboratory). The wild-type ($Ash1L^{+/+}$), heterozygous *Ash1L*-KO ($Ash1L^{+/1f}$), and homozygous *Ash1L*-KO ($Ash1L^{1f/1f}$) mice were generated by $Ash1L^{+/1f}$ x $Ash1L^{+/1f}$ mating. (2) Generating *Ash1L*-Nestin-cKO mice: The *Ash1L* neural conditional knockout mice were generated by mating *Ash1L* floxed mice with Nestin-cre mice (B6.Cg-Tg (Nes-cre) 1Kln/J, The Jackson Laboratory). The wild-type ($Ash1L^{2f/2f};Nestin-Cre^{-/-}$), heterozygous ($Ash1L^{2f/+};Nestin-Cre^{+/-}$), and homozygous *Ash1L*-Nes-cKO ($Ash1L$-Nes-cKO, $Ash1L^{2f/2f};Nestin-Cre^{+/-}$) were generated by $Ash1L^{2f/2f};Nestin-Cre^{-/-}$ (female) x $Ash1L^{2f/+};Nestin-Cre^{+/-}$ (male) mating. (3) The 4-hydrotamoxifen inducible *Ash1L*-cKO mice were generated by $Ash1L^{+/2f}$; Rosa26-CreER$^{T2+/-}$ x $Ash1L^{+/2f}$; Rosa26-CreER$^{T2+/-}$ mating. The Rosa26-CreER$^{T2}$ mouse line (B6.129-Gt(ROSA)26Sor$^{tm1(cre/ERT2)Tyj}$/J) was purchased from the Jackson Laboratory.

**Genotyping**. Genomic DNA was extracted from mouse tails with lysis buffer of 0.01 M NaOH. After neutralization with Tris-HCl (PH 7.6), the extracted genomic DNA was used for genotyping PCR assays. Primers used for genotyping were listed in Supplementary Table 3.

**Isolation, culture and induced differentiation of neural progenitor cells**. The neural tissues were isolated from the subventricular zone of P30 brains and dissociated with 0.05% trypsin-EDTA (Life Technologies) at 37 °C for 15 min. The reaction was stopped by trypsin inhibitor (10 mg/ml, Worthington Biochemical Corporation). The dissociated cells were washed with cold PBS for 3 times and plated onto the non-coated petri dishes in the Neurobasal medium (Life Technologies) supplemented with 1× B27 supplement (Gibco), 1x GlutaMAX (Life Technologies), 20 ng/ml murine epithelial growth factor (Peprotech), 20 ng/ml basic fibroblast growth factor (Peprotech), and 100U/ml penicillin/streptomycin (Life Technologies). 5–7 days later, the neurospheres formed by proliferating NPCs were collected and re-plated onto the Poly-L-Ornithine (R&D systems) and Laminin (Corning)-coated plates to form monolayer culture. The neural progenitor cells were passaged at 1:5 ratio every 3 days. To induce NPCs differentiation, the NPC monolayer cells were gently washed with PBS for 3 times and cultured under the Neurobasal medium supplemented with 1× N2 supplement (Giboco), 1x GlutaMAX (Life Technologies).

**Induced *Ash1L* deletion in neural progenitor cells**. *Ash1L* gene deletion in the $Ash1L^{2f/2f}$;Rosa26-CreER$^{T2+/-}$ NPCs were induced by the addition of 4-hydroxytamoxifen (Sigma-Aldrich) at 0.1 μM in the culture medium. Genotyping was used to confirm the *Ash1L* gene deletion 10 days after 4-hydrotamoxifen treatment. The minimize potential effects of 4-hydroxytamoxifen on gene expression, the confirmed *Ash1L*-deleted NPCs were further cultured in the NPC culture medium without 4-hydroxytamoxifen for three passages before further experiments.

**Alizarin red/alcian blue bone staining**. Whole mouse carcasses were collected after euthanasia, defatted for 2–3 days in acetone, stained sequentially with Alcian blue and alizarin red S in 2% KOH, cleared with 1% KOH/20% glycerol, and stored in 50% EtOH/50% glycerol.

**Microcomputed tomography (micro-CT)**. Skulls were serially imaged using a PerkinElmer Quantum GX micro-CT scanner. The following image acquisition parameters were used at each scan time point: 4 min acquisition; 90 kVp/88 μA; Field of View (FOV), 45 mm; pixel resolution, 90 μm. Then, tissue sections were collected for histology and Ta analysis using ICP-OES.

**Nissl staining**. The paraffin blocks were prepared by Division of Human Pathology of Michigan State University. Briefly, mouse brain sections were dewaxed in xylene and rehydrated in alcohol. Then, sections were stained in toluidine buffer [1 g toluidine blue (Sigma) in 100 mL 95% ethanol] at room temperature for 20 min. Quick rinse in tap water and 70% ethanol to remove excess stain. After dehydration and wax, sections were mounted in mounting media H5000 (Vector Laboratories). Images were captured using a Zeiss Axio Imager microscope (Carl Zeiss GmbH, Oberkochen, Germany) and an installed AxioCam HRc camera (Carl Zeiss GmbH) with image acquisition via Zeiss Zen Pro software (v.2.3; Carl Zeiss GmbH).

**Immunostaining**. Mouse tissues were fixed in 4% PFA in PBS overnight at 4 °C and embedded in paraffin. For immunofluorescence, tissue sections of 5 μm were cut, dewaxed and rehydrated. Antigen retrieval was performed by microwaving the sections on 0.01 M sodium citrate buffer (pH 6.0) for 4 min. Tissue sections were blocked in 5% normal donkey serum (NDS) for 30 min after sensing with PBS. Tissue sections then were incubated with primary antibodies diluted in 5% NDS overnight at 4 °C. Antibodies used were: mouse anti-SATB2 (1:10, ab51502, abcam), rat anti-CTIP2 (1:100, ab18465, abcam), and rabbit anti-MBP (1:500;

78896; Cell Signaling technology). After washing with PBS, sections were incubated with Alexa Fluor 488 donkey anti-mouse IgG (1:300; 715-545-150; Jackson ImmunoResearch) or R-Phycoerythrin AffiniPure F(ab')₂ Fragment Donkey Anti-Rat IgG (1:300; 712-116-153; Jackson ImmunoResearch) for 1 h and mounted using Vectorshield mounting media with DAPI (H1200, Vector Laboratories). Images were captured using a Zeiss Axio Imager microscope (Carl Zeiss GmbH, Oberkochen, Germany) and an installed AxioCam HRc camera (Carl Zeiss GmbH) with image acquisition via Zeiss Zen Pro software (v.2.3; Carl Zeiss GmbH).

**ASH1L antibody generation**. The Polyclonal Rabbit anti-ASH1L antibody was generated by Pocono Rabbit Farm & Laboratory. The recombinant mouse ASH1L peptides (aa 2053-2347) were used as antigen. The antibodies were purified by immunoaffinity chromatography using antigen-coupled affi-gel 10 (Bio-rad).

**Western Blot analysis**. Total proteins were extracted by RIPA buffer and separated by electrophoresis by 8–10% PAGE gel. The protein was transferred to the nitrocellulose membrane and blotted with primary antibodies. The antibodies used for Western Blot and IP-Western Blot analyses included: rabbit anti-Ash1L (1:1000, in house) and IRDye 680 donkey anti-rabbit second antibody (1: 10000, Li-Cor). The images were developed by Odyssey Li-Cor Imager (Li-Cor).

**RNA extraction and qRT-PCR assays**. Acute slices of E16.5 cortices were homogenized in the TRI Reagent (Sigma) in a Dounce homogenizer. 1-bromo-3-choropropane (Sigma) was added to the homogenized tissues, followed by centrifugation to separate the phases. The RNA-containing phase was mixed with isopropanol and the total RNA was precipitated by centrifugation. Total RNA was extracted from cells by QI shredder (Qiagen) and RNeasy mini purification kit (Qiagen). Total RNA (1 μg) was subjected to reverse transcription using Iscript reverse transcription supermix (Bio-Rad). cDNA levels were assayed by real-time PCR using iTaq universal SYBR green supermix (Bio-Rad) and detected by CFX386 Touch Real-Time PCR detection system (Bio-Rad). Primer sequences for qPCR are listed in Supplementary Table 3.

**RNA-seq sample preparation for HiSeq4000 sequencing**. Total RNA (1 μg) was used to generate RNA-seq library using NEBNext Ultra Directional RNA library Prep Kit for Illumina (New England BioLabs, Inc) according to the manufacturer's instructions. Adapter-ligated cDNA was amplified by PCR and followed by size selection using agarose gel electrophoresis. The DNA was purified using Qiaquick gel extraction kit (Qiagen) and quantified both with an Agilent Bioanalyzer and Invitrogen Qubit. The libraries were diluted to a working concentration of 10 nM prior to sequencing. Sequencing on an Illumina HiSeq4000 instrument was carried out by the Genomics Core Facility at Michigan State University.

**RNA-Seq data analysis**. RNA-Seq data analysis was performed essentially as described previously. All sequencing reads were mapped mm9 of the mouse genome using Tophat2[32]. The mapped reads were normalized to reads as Reads Per Kilobase of transcript per Million mapped reads (RPKM). The differential gene expression was calculated by Cuffdiff program and the statistic cutoff for identification of differential gene expression is $p < 0.01$ and 1.5-fold RPKM change between samples. The heatmap and plot of gene expression were generated using plotHeatmap and plotProfile in the deepTools program[33]. The differential expressed gene lists were input into the GENEONTOLOGY for the GO enrichment analyses (http://geneontology.org/).

**Behavioral tests**. All behavioral tests were performed on littermates of wild-type and homozygous Ash1L-Nes-cKO mice. The mice were labeled by ear-tags and passed to AJ Robison's lab for the behavioral tests. All the behavioral tests were performed by Dr. Natalia Duque-Wilckens who was blinded to the genotypes of animals during the behavioral tests.

**Open field test**. The open-field apparatus consisted of a custom-made, square white polyvinylchloride foam box (38 × 38 × 35 cm). Their behavior was recorded for the first 10 min of habituation to measure time spent in open field, time spent in corners, and time freezing with a digital CCD camera connected to a computer running an automated video tracking software package (Clever Sys).

**Novel object recognition test (NOR)**. NOR was assessed using a 3-day paradigm that included habituation, training, and testing as described previously[34–37]. Each day, mice were acclimated for 60 min to the behavioral testing room before assessment. All tests were performed under red lights, and behaviors were video recorded and automatically scored using Clever Sys. During habituation (day1), mice were placed into the open field apparatus, a square white polyvinylchloride foam 38 × 38 × 35 cm box, for 60 min while video recorded. The first 5 and 10 min were assessed for locomotor behavior in the open field (Fig. 3n). For training (day 2), two identical objects consisting of miniature wheels, knobs, spark plugs, and

Lego blocks were placed in opposite corners of the open field apparatus, and the animals were allowed to explore the objects for 60 min. The object pairs used were counterbalanced across treatments. For testing (day 3), mice were placed in the same apparatus, but this time one object of the pair was replaced with another dissimilar object (novel object), and they were allowed to freely explore for 5 min. Their behavior was recorded, and the time the mice spent with their nose oriented towards the object within 3.5 cm of the object edge was considered exploration time. Throughout testing, objects and apparatus were cleaned with 70% ethanol between trials. Discrimination index was calculated as

$$\text{DI} = \frac{(\text{time investigating novel} - \text{time investigating familiar})}{(\text{time investigating novel} + \text{time investigating familiar})} \quad (1)$$

**Sociability and preference for social novelty test**. This test was adapted from Crawley's sociability and preference for social novelty protocol[16,17], which consists of three phases. Mice were acclimated for 60 min to the behavioral testing room under red lights before testing. The behaviors during all three phases were video recorded and automatically scored using Clever Sys. In phase I (habituation), the experimental mouse was placed in the center of a three-chamber apparatus (polyvinylchloride, 60 × 40 × 22 cm, Fig. 3a, c) and allowed to freely explore for 5 min. During this time, the mouse had free access to all three chambers, which are connected by small openings at the bottom of the dividers. In phase 2 (sociability), two identical, wire cup-like containers were placed one in each of the side chambers. In this phase, an unfamiliar same-sex mouse was placed in one of the containers ("social stimulus"), while the other remained empty ("object"). The experimental mouse was allowed to freely explore the three chambers again for 5 min. In phase 3 (social memory), the container with the mouse (now "known") was moved to the opposite chamber, and a new same-sex mouse ("unknown") was placed in the other container. The experimental mouse was allowed to freely explore the three chambers for 5 min. Throughout testing, objects and apparatus were cleaned with 70% ethanol between trials. For analysis, the time with total body spent in each of the three chambers was recorded.

**Quantitative self-grooming measurement**. Mice were acclimated for 60 min to the behavioral testing room before assessment. After the mice were placed into an open field apparatus (40 × 40 × 40 cm) and habituated for 30 min, the mouse grooming behavior was video recorded for 10 min. The total grooming time and number of grooming episodes were manually measured.

**Paw clasping scoring**. The paw clasping scoring method was adopted from the reported protocol[38]. Briefly, the mice were suspended by tail for 10 s. If the hindlimbs are consistently splayed outward, away from the abdomen, it was assigned a score of 0. If both hindlimbs were partially retracted toward the abdomen for more than 50% of the time suspended, it was designated as mild paw clasping and received a score of 1. If its hindlimbs were entirely retracted and touching the abdomen for more than 50% of the time suspended, it was designated as severe paw clasping and received a score of 2.

**Statistics and reproducibility**. All statistical analyses were performed using GraphPad Prism 8 (GraphPad Software). Parametric data were analyzed by a two-tailed t test or two-way ANOVA test for comparisons of multiple samples. P-values < 0.05 were considered statistically significant. Planned comparisons (Šídák's multiple comparisons test) were used if ANOVAS showed significant main or interaction effects. Data are presented as mean ± SEM.

**Reporting summary**. Further information on research design is available in the Nature Research Reporting Summary linked to this article.

## Data availability
Source data underlying the main figures are presented in Supplementary Data 3. The RNA-seq data presented in this study has been deposited to the Gene Expression Omnibus database, GEO accession: GSE173262. Other data have been disclosed in the sections above, or are available from the corresponding author upon reasonable request.

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

## Acknowledgements

The authors thank Drs. Jeremy Hix, Christiane Mallett, and Erik Shapiro for micro-CT scan imaging assays. MSU genomics core facility processed the next-generation sequencing. This work was supported by the National Institutes of Health (grant R01GM127431).

## Author contributions

J.H. conceived the project. Y.G., N.D., J.H. performed the experiments. Y.G. and M.B.A. maintained the mouse colonies. Y.W. prepared the recombinant protein for the Ash1L antibody generation. A.J.R. oversaw the mouse behavior tests. G.I.M. and J.H. performed the sequencing data analysis. Y.G., N.D., A.J.M., A.J.R., and J.H. interpreted the data. Y.G., N.D., and J.H. wrote the manuscript. Y.G. and N.D. contribute equally to this work.

## Competing interests

The authors declare no competing interests.
