## [Peer Review File · Communications Biology]

Reviewers' comments:

Reviewer #1 (Remarks to the Author):

This interesting manuscript investigates phenotypes of a new conditional line of mice with a mutation in the histone methyltransferase gene *Ash1l*. Assays evaluated craniofacial deformity, neuroanatomical and myelination abnormalities, gene expression profiles, and behaviors relevant to diagnostic symptoms of autism spectrum disorder and intellectual disability, congruent with behavioral outcomes reported in humans with an *ASH1L* mutation.

While the behavioral methods and results are generally well presented, and appropriate Ns were used, several key points of information are missing.

1. Statistical results must be fully stated, including F values and degrees of freedom for comparisons using ANOVA, and t values and degrees of freedom for comparisons using Student's t test, within the figure legend of each graph.

2. Scoring procedures for the novel object recognition and 3-chambered social approach scores must be described. If automated Cleversys videotracking was used, please describe the programmed scoring parameters. If human observers conducted the scoring, in real time or from videos, please describe the scoring protocols.

3. What was the breeding strategy? Were WT and knockout mice littermates?

4. Please state the methods used to keep the experimenters blind to genotype during behavioral testing.

5. Quantitation is needed for the overgrooming and forepaw clasping assays. Figure 3 panels E and F are photographs of skin lesions in one knockout mouse. Movies S1 and S2 similarly are illustrations of a single case of forepaw movements in one wildtype and one knockout respectively.

Please conduct a standard test to quantify self-grooming on the full set of mice of each genotype, with appropriate statistical analysis to evaluate genotype differences. Similarly, please use rating scales in the literature to score forepaw grasping in the full Ns of each genotype, followed by statistical analysis.

If sufficient mice are no longer available for these additional behavioral assays, statements about repetitive behaviors must be removed throughout.

6. Both novel object recognition and 3-chambered social approach results could be caused by general hyperactivity of the mutant line. The authors correctly conducted a control for general locomotion. Total distance traveled was measured during the day 1 habituation session preceding the novel object recognition test. Figure 3 Panel I indicates higher total distance in the mutants as compared to wildtype during the habituation session.

Panel I appears to represent the sum of total activity during the 60 minute habituation session. In contrast, the novel object recognition and social approach test sessions were of 5 or 10 minute durations. Can the authors display the 60 minute test session data as 5 minute time bins across the 60 minute session? Two-Way ANOVA with an appropriate post-hoc test such as Dunnett's or Newman-Keuls would then be employed to compare total distance scores between genotypes during the first 5 minutes, and during the first 10 minutes. This data presentation will provide a more direct comparison to the test session lengths of the novel object recognition and 3-chambered assays.

If these analyses reveal considerably higher activity by the *Ash1l* group as compared to wildtypes, then elevated general locomotion could be the reason the knockouts did not focus their attention on the social partner, on the novel object, and on the center of the open field. This alternative hypothesis must be clearly stated. Interpretation of the behavioral results would then require

considerably more caution about relevance to autism.

7. Please proofread again to correct typographical errors. As examples:

Page 7 "implicate Ash1L might also play important roles" is grammatically incorrect

Page 5: "we first focused on testing these two core autistic behaviors" should read "autism-relevant" or "autism-like," as used elsewhere in the manuscript

References to the behavioral testing methods do not appear to match the citation numbers. E.g. reference numbers 16 and 33 do not cite original papers detailing standardized methods for these assays.

Reviewer #2 (Remarks to the Author):

Mutations of the epigenetic gene ASH1L, a histone methyltransferase, have been identified in individuals with ASD and intellectual disability (ID). This study generated a mouse model with ASH1L cKO to test whether it may be sufficient to cause these phenotypes. They show that the global KO is lethal shortly after birth. Mice with ASH1L cKO in the brain, using neural progenitor cell (NPC)-specific Cre (Nestin-Cre), show ASD relevant behaviors and cognitive deficits. To begin to identify the underlying molecular mechanisms, RNASeq of cultured NPCs with ASH1L cKO revealed several gene expression level changes, both up- and down- regulation. Several of the downregulated genes link to brain development based on enriched gene ontology analyses.

The strength of this work is the data supporting a causative role for ASH1L gene mutations in ASD and ID phenotypes.

However, there are also some moderate weaknesses.

- It is not specified whether the cKO data is from homozygotes.
- It is important to know whether the heterozygotes show a phenotype as this is more representative of the human disorder.
- It should be stated, if known, whether ASH1L is expressed in all cell and tissue types, as this relates to the multiple diverse symptoms seen in humans.
- At a minimum, a subset of the selected 7 downregulated genes should be tested for similar changes in the in vivo cKOs. This is important because the RNASeq was performed on isolated NPCs, maintained in vitro for many days, passaged multiple times, and induced to differentiate. These manipulations may have affected the gene expression changes.
- The up regulated genes are not discussed, beyond showing the enriched GO pathways in the supplement.

Minor comments:

- Does the RNASeq data suggest why Satb2 positive neuron migration may be selectively altered in the cKO (Fig. 2)?
- Why was lamination only examined in the prefrontal cortex?

Reviewer #3 (Remarks to the Author):

In the manuscript entitled "Loss of histone methyltransferase ASH1L in developing brains causes autistic-like behaviors in a mouse model", Gao and his colleagues studied the brain deficits after genetic ablation of Ash1L during brain development. The authors provided convinced evidences showing malformations of cortical formation, delayed myelination, and abnormal social, cognitive and anxiety behaviors in Ash1L Nes-cKO mice, which were consistent with some of the clinical manifestations in autism patients carrying ASH1L mutation. Overall, both the cellular and

behavioral data are very convincing. However, I am quite surprised to see the subtle change of transcriptome in cultured KO neurons, which is somehow not very consistent with the observed severe phenotype. And also, I am not sure why for the RNA-seq data, the authors didn't do direct comparison between KO v.s. WT at different time points. Also, as mentioned in the manuscript, the authors intended to minimize the secondary effects caused by the mixed cell types. To this end, instead of using induced differentiated neuronal cells in vitro, simply check the transcriptome of NPCs between KO and WT may give more clue for genes directly regulated by ASH1L. RNA-seq using bulk cortical tissue at the critical timing (P14) for synaptic plasticity is also recommended.

And below are two minor comments:

1. For the data related to Figure 2B, the authors observed abnormal location of SATB2+ neurons in Ash1L-KO cortex, and concluded it was due to impaired migration, while lack of further supporting evidence. Such statement might be okay for the discussion, but not in the results section.

2. Although clamping and over-grooming are considered as autism-like behaviors, these behaviors can also be observed in control animals or animals with other conditions. In the manuscript, the authors only provided this information from KOs. Although it might be difficult to provide statistical results, it will be worth to mention the condition of controls as well.

Response to reviewers:

We would like to thank the reviewers' constructive comments for our manuscript, which largely improves the overall quality of this study. We have taken the reviewers' comments into careful consideration and added new data to address reviewers' comments in the revised manuscript. The followings are the point-by-point responses to reviewer's comments:

Reviewer #1 (Remarks to the Author):

This interesting manuscript investigates phenotypes of a new conditional line of mice with a mutation in the histone methyltransferase gene *Ash1l*. Assays evaluated craniofacial deformity, neuroanatomical and myelination abnormalities, gene expression profiles, and behaviors relevant to diagnostic symptoms of autism spectrum disorder and intellectual disability, congruent with behavioral outcomes reported in humans with an *ASH1L* mutation.

We thank the reviewer's positive comments on our work.

While the behavioral methods and results are generally well presented, and appropriate *Ns* were used, several key points of information are missing.

1. Statistical results must be fully stated, including *F* values and degrees of freedom for comparisons using ANOVA, and *t* values and degrees of freedom for comparisons using Student's *t* test, within the figure legend of each graph.

Response: We apologize for this omission. The manuscript now contains full statistical parameters for each result described. We used the convention of (*FX,NN* = *N.NNN*, *p* < *N.NNN*) for reporting ANOVA and (*t* = *N.NNN*, *df* = *NN*, *p* < *N.NNN*) for reporting *t* tests.

2. Scoring procedures for the novel object recognition and 3-chambered social approach scores must be described. If automated Cleversys videotracking was used, please describe the programmed scoring parameters. If human observers conducted the scoring, in real time or from videos, please describe the scoring protocols.

Response: We have added detailed methodology for behavioral scoring. In brief, behavior was scored automatically by CleverSys software, and we now describe the size and shape of each apparatus and the regions defined for automated scoring.

3. What was the breeding strategy? Were WT and knockout mice littermates?

Response: We have added the detailed mouse breeding strategy in the Methods (page 10) and specified that "All behavioral tests were performed on the littermates of wild-type and homozygous *Ash1L*-Nes-cKO mice" in the "Behavioral tests" of Materials and Methods section (page 13).

4. Please state the methods used to keep the experimenters blind to genotype during behavioral testing.

Response: We specified “The mice were labeled by ear-tag and passed to AJ Robison’s lab for the behavioral tests. All the behavioral tests were performed by Dr. Natalia Duque-Wilckens who was blinded to the genotypes of animals during the experiments.” In the “Behavioral tests” of Materials and Methods section (page 13).

5. Quantitation is needed for the overgrooming and forepaw clasping assays. Figure 3 panels E and F are photographs of skin lesions in one knockout mouse. Movies S1 and S2 similarly are illustrations of a single case of forepaw movements in one wildtype and one knockout respectively.

Please conduct a standard test to quantify self-grooming on the full set of mice of each genotype, with appropriate statistical analysis to evaluate genotype differences. Similarly, please use rating scales in the literature to score forepaw grasping in the full Ns of each genotype, followed by statistical analysis.

If sufficient mice are no longer available for these additional behavioral assays, statements about repetitive behaviors must be removed throughout.

Reponses: Thanks for pointing this out. We added to the detailed methods for the quantitative measurement of grooming and paw clasping (Methods: “self-grooming measurement” and “paw clasping scoring” page 15), followed by statistical analysis. The results are shown in the figures 3E-I. Videos showing hair grooming, mild and severe paw clasping of *Ash1L*-cKO mice were added as the supplemental videos (Movie S3-S5).

6. Both novel object recognition and 3-chambered social approach results could be caused by general hyperactivity of the mutant line. The authors correctly conducted a control for general locomotion. Total distance traveled was measured during the day 1 habituation session preceding the novel object recognition test. Figure 3 Panel I indicates higher total distance in the mutants as compared to wildtype during the habituation session.

Panel I appears to represent the sum of total activity during the 60 minute habituation session. In contrast, the novel object recognition and social approach test sessions were of 5 or 10 minute durations. Can the authors display the 60 minute test session data as 5 minute time bins across the 60 minute session? Two-Way ANOVA with an appropriate post-hoc test such as Dunnett’s or Newman-Keuls would then be employed to compare total distance scores between genotypes during the first 5 minutes, and during the first 10 minutes. This data presentation will provide a more direct comparison to the test session lengths of the novel object recognition and 3-chambered assays.

If these analyses reveal considerably higher activity by the *Ash1l* group as compared to wildtypes, then elevated general locomotion could be the reason the knockouts did not focus their attention on the social partner, on the novel object, and on the center of the open field. This alternative hypothesis must be clearly stated. Interpretation of the behavioral results would then require considerably more caution about relevance to autism.

Response: This is a very important point, and we thank the reviewer for pointing it out. When we analyze only the first five minutes of open field locomotor activity, we see no difference between genotypes (New figure 3N). This indicates that the differences we report in our 5-minute sociability, social memory, and novel object recognition assays are not simply driven by hyperactivity. However, with longer observation (10 minutes in the open field, which is what our original graph reported), we do see a hyperactivity phenotype in the KO mice (New figure 3N). We apologize for the lack of clarity in the time of each assay, and we now try to make the times very clear in the methods section.

7. Please proofread again to correct typographical errors. As examples:

Page 7 “implicate Ash1L might also play important roles” is grammatically incorrect

Response: We thank the reviewer for pointing out the error, and “implicate” has now been changed to “suggest that” (page 8).

Page 5: “we first focused on testing these two core autistic behaviors” should read “autism-relevant” or “autism-like,” as used elsewhere in the manuscript

Response: We thank the reviewer for pointing out that error. We changed the “autistic behaviors” to “autism-like behaviors” (page 5).

References to the behavioral testing methods do not appear to match the citation numbers. E.g. reference numbers 16 and 33 do not cite original papers detailing standardized methods for these assays.

Responses: We cited the original papers (ref. 17, 36-39) detailing standardized methods for the behavioral tests.

Reviewer #2 (Remarks to the Author):

Mutations of the epigenetic gene ASH1L, a histone methyltransferase, have been identified in individuals with ASD and intellectual disability (ID). This study generated a mouse model with ASH1L cKO to test whether it may be sufficient to cause these phenotypes. They show that the global KO is lethal shortly after birth. Mice with ASH1L cKO in the brain, using neural progenitor cell (NPC)-specific Cre (Nestin-Cre), show ASD relevant behaviors and cognitive deficits. To begin to identify the underlying molecular mechanisms, RNASeq of cultured NPCs with ASH1L cKO revealed several gene expression level changes, both up- and down- regulation. Several of the downregulated genes link to brain development based on enriched gene ontology analyses.

The strength of this work is the data supporting a causative role for ASH1L gene mutations in ASD and ID phenotypes.

We thank the reviewer’s positive comments on our work.

However, there are also some moderate weaknesses.

- It is not specified whether the cKO data is from homozygotes.

Reponses: We specify that only homozygous cKO mice were used for the experiments in both Results and Methods sections (page 13).

- It is important to know whether the heterozygotes show a phenotype as this is more representative of the human disorder.

Reponses: We thank the reviewer pointing this out. We agree with the reviewer that heterozygous *Ash1L* disruptive mutation is more representative of situations of human patients. Considering the primary aim of current study was to prove the causality between *Ash1L* loss in the developing brain and genesis of ASD/ID, we did not include the heterozygous *Ash1L*-KO mice to quantitatively measure whether the degree of ASD/ID-like defects was correlated with *Ash1L* gene dose in the developing brain. We discuss this as one of limitations for the current study in the Discussion section (page 9).

- It should be stated, if known, whether ASH1L is expressed in all cell and tissue types, as this relates to the multiple diverse symptoms seen in humans.

Response: We searched the key literatures and found the mouse ENCODE transcriptome data about the *Ash1L* expression levels in various mouse tissues (attached figure, not shown in the manuscript). We stated in the text “The mouse ENCODE transcriptome data showed that *Ash1L* is highly expressed in the embryonic and adult brain, suggesting its possible role in brain development and function” (page 6).

- At a minimum, a subset of the selected 7 downregulated genes should be tested for similar changes in the *in vivo* cKOs. This is important because the RNASeq was performed on isolated NPCs, maintained *in vitro* for many days, passaged multiple times, and induced to differentiate. These manipulations may have affected the gene expression changes.

Responses: ASH1L is an epigenetic factor regulating lineage-specific gene expression during normal development. To examine the function of ASH1L in regulating lineage-specific gene expression during brain development, we intended to examine the dynamic gene expression changes during NPC differentiation as well as to identify the differentially expressed genes between WT and KO in specific cell lineages that are located at the same developmental stages. However, the bulk developing brain contain interconnected heterogenous neural lineages or even the same lineages with different developmental stages. In addition, the gene expression changes quickly during early NPC differentiation (our RNA-seq data showed the gene expression changed dramatically 24 hours after induced differentiation *in vitro*, Fig. S3H, S3I), which made it difficult to do the analysis on WT and KO developing brains with precisely matched developmental time points. For these reasons, the validation of differentially expressed gene *in vivo* by dynamically and quantitatively measuring the mRNA or protein levels in specific neural lineages as well as at specific brain developing time points is technically difficult.

To overcome this technical obstacle, in this study we chose to delete *Ash1L* in a homogenous NPC population and compare the gene expression with its wild-type parental cell population *in vitro*. In the revised manuscript, we added the qRT-PCR results (new figure 4E-K) to confirm the seven differentially expressed genes (Fig. 4D) identified by RNA-seq. To minimize the potential effects of 4OH-TAM on the gene expression, we cultured the NPCs in the NPC medium without 4OH-TAM for three passages before induced differentiation and gene expression analysis (Materials and Methods, page 11).

Finally, we strongly agree with the reviewer that the validation of differential gene expression *in vivo* is critical for understanding the molecular mechanisms directly linking to the function of ASH1L in brain development and pathogenesis of ASD/ID. In the next stage of study, we will combine the neural lineage-specific reporter and neural lineage-specific Cre mouse lines to isolate different neural lineages at various brain developmental time points for the gene expression analysis. We discuss this as one of the limitations for current study in the Discussion section (page 9).

- The up regulated genes are not discussed, beyond showing the enriched GO pathways in the supplement.

Responses: We apologize for the lack of clarity in explaining how we did the transcriptome analysis for this study, and we now try to clarify it in the text. In this study, we first examined the dynamic transcriptome changes in wild-type NPCs to identify the genes that had changed expression during normal NPC differentiation. The results identified total 2,475 upregulated and 2,808 downregulated genes during induced differentiation of wild-type NPCs (Figure S3H-I). The GO enrichment analysis of these up- and down-regulated “reflects the dynamic neural lineage development, cell cycle exit, and reduced metabolism during NPC differentiation” (added on page 7). Since ASH1L functions as an epigenetic factor facilitating transcriptional activation, we focused on the 2,475 upregulated genes and set to identify the genes that had impaired expression in the *Ash1L*-KO cells (page 7). Among the 2,475 genes upregulated in the differentiating wild-type NPCs, 70 genes were found to have significantly reduced expression in the *Ash1L*-KO cells (Fig. 4A). The GO enrichment analysis showed the enriched GO terms of these 70 genes were involved in telencephalon development, regulation of cell

communication, brain development, and central nervous development.

Minor comments:

- Does the RNASeq data suggest why Satb2 positive neuron migration may be selectively altered in the cKO (Fig. 2)?

Response: A previous study reported that loss of *Emx2* impaired the neuronal migration in the developing mouse cortex (ref. 30), which might cause the delayed cortical lamination observed in the *Ash1L*-KO developing cortex. We added this point in the Discussion section (page 8).

- Why was lamination only examined in the prefrontal cortex?

Response: We examined the multiple cortical areas in the histological studies. Therefore, in the text we changed the FPC to cortex or cortices (page 4).

Reviewer #3 (Remarks to the Author):

In the manuscript entitled “Loss of histone methyltransferase ASH1L in developing brains causes autistic-like behaviors in a mouse model”, Gao and his colleagues studied the brain deficits after genetic ablation of Ash1L during brain development. The authors provided convinced evidences showing malformations of cortical formation, delayed myelination, and abnormal social, cognitive and anxiety behaviors in Ash1L Nes-cKO mice, which were consistent with some of the clinical manifestations in autism patients carrying ASH1L mutation. Overall, both the cellular and behavioral data are very convincing. However, I am quite surprised to see the subtle change of transcriptome in cultured KO neurons, which is somehow not very consistent with the observed severe phenotype. And also, I am not sure why for the RNA-seq data, the authors didn't do direct comparison between KO v.s. WT at different time points. Also, as mentioned in the manuscript, the authors indented to minimize the secondary effects caused by the mixed cell types. To this end, instead of using induced differentiated neuronal cells in vitro, simplify check the transcriptome of NPCs between KO and WT may give more clue for genes directly regulated by ASH1L. RNA-seq using bulk cortical tissue at the critical timing (P14) for synaptic plasticity is also recommended.

Response: We thank the reviewer's positive comments on our work.

(1) Subtle change of transcriptome: We apologize for the lack of clarity in explaining how we did the transcriptome analysis for this study, and we now try to clarify it in the text. In this study, we first examined the dynamic transcriptome changes in wild-type NPCs to identify the genes that had changed expression during normal NPC differentiation. The results identified total 2,475 upregulated and 2,808 downregulated genes during induced differentiation of wild-type NPCs (Figure S3H-I). Since ASH1L functions as an epigenetic factor facilitating transcriptional activation, we focused on the 2,475 upregulated genes and set to identify the genes that had impaired expression in the *Ash1L*-KO cells (page 7). Among the 2,475 genes upregulated in the differentiating wild-type NPCs, 70 genes were found to have significantly reduced expression in the *Ash1L*-KO cells (Fig. 4A). In addition, we only examined the transcriptome changes 24 hours after induced differentiation, which might reflect the limited number of genes

that had impaired expression in the *Ash1L*-KO cells during very early stage of NPC differentiation.

(2) Check the transcriptome of NPCs between KO and WT and RNA-seq using bulk cortical tissue at the critical timing: ASH1L is an epigenetic factor regulating lineage-specific gene expression during normal development. To examine the function of ASH1L in regulating lineage-specific gene expression during brain development, we intended to examine the dynamic gene expression changes during NPC differentiation as well as to identify the differentially expressed genes between WT and KO in specific cell lineages that are located at the same developmental stages. However, the bulk developing brain contain interconnected heterogenous neural lineages or even the same lineages with different developmental stages. In addition, the gene expression changes quickly during early NPC differentiation (our RNA-seq data showed the gene expression changed dramatically 24 hours after induced differentiation *in vitro*, Fig. S3H, S3I), which made it difficult to do the analysis on WT and KO developing brains with precisely matched developmental time points. We have tried to do the RNA-seq analysis on the bulk brain tissues. However, it was very hard to interpret whether the differential gene expression was caused by *Ash1L* deletion or other variables such as mixed cell types in the samples or the cells located at different developmental stages. For the same reason, we anticipate that the RNA-seq analysis on the P14 bulk cortical tissues will encounter the same technical difficulties.

To overcome this technical obstacle, in this study we chose to delete *Ash1L* in a homogenous NPC population and compare the gene expression with its wild-type parental cell population *in vitro*. In the revised manuscript, we added the qRT-PCR results (new figure 4E-K) to confirm the seven differentially expressed genes (Fig. 4D) identified by RNA-seq. To minimize the potential effects of 4OH-TAM on the gene expression, we cultured the NPCs in the NPC medium without 4OH-TAM for three passages before induced differentiation and gene expression analysis (Materials and Methods, page 11).

Finally, we strongly agree with the reviewer that the validation of differential gene expression *in vivo* is critical for understanding the molecular mechanisms directly linking to the function of ASH1L in brain development and pathogenesis of ASD/ID. In the next stage of study, we will combine the neural lineage-specific reporter and neural lineage-specific Cre mouse lines to isolate different neural lineages at various brain developmental times for the gene expression analysis. We discuss this as one of the limitations for current study in the Discussion section (page 9).

And below are two minor comments:

1. For the data related to Figure 2B, the authors observed abnormal location of SATB2⁺ neurons in *Ash1L*-KO cortex, and concluded it was due to impaired migration, while lack of further supporting evidence. Such statement might be okay for the discussion, but not in the results session.

Response: We thank the reviewer for pointing out this. We changed the text to “In contrast, some SATB2⁺ neurons in the *Ash1L*-KO FC did not properly located in the upper layers and scattered in the bottom layers”.

2. Although clamping and over-grooming are considered as autism-like behaviors, these behaviors can also be observed in control animals or animals with other conditions. In the

manuscript, the authors only provided this information from KOs. Although it might be difficult to provide statistical results, it will be worth to mention the condition of controls as well.

Reponses: We thank the reviewer for pointing this out. We added to the detailed methods for the quantitative measurement of grooming and paw clasping (Methods: “self-grooming measurement” and “paw clasping scoring” page 15), followed by statistical analysis. The results are shown in the figures 3E-I. Videos showing hair grooming, mild and severe paw clasping of *Ash1L*-cKO mice were added as the supplemental videos (Movie S3-S5).

All revised parts in the manuscripts are highlighted as **dark red**.

Reviewers' comments:

Reviewer #1 (Remarks to the Author):

The authors have effectively addressed all previous concerns. They are to be commended for their thoughtful replies, and for the comprehensive changes made.

Reviewer #2 (Remarks to the Author):

The revised manuscript is improved by the newly added statistics and details requested. However, not included in the revision are approaches that address two concerns that were raised in the first review. This reduces the significance of the study. The concerns are highly relevant to the fundamental question stated by the authors. What are the molecular mechanisms linking Ash1L mutations to the pathogenesis of ASD/ID?

1. It was recommended that heterozygotes be tested for ASD and ID phenotypes, rather than only homozygote Ash1-L KO, to increase the relevance to the human disorder. However, this was not included in the revision.

2. There are no tests in vivo for any of the molecular changes identified here in cultured NPCs. The NPCs were induced to differentiate in vitro and assessed by RNA seq 24 hrs later. Seventy genes were identified that are not upregulated in Ash1L KO relative to WT NPCs. It is important to verify in vivo relevance of at least some of the molecular changes found in vitro. In response to the review, the authors feel it is necessary to only test NPCs in vivo and this is prohibitive because of mixed cell types and maturation stages in vivo. However, it seems likely that at least some of the gene expression changes should be detectable at older ages, because: Ash1L is expressed throughout development with high levels in the adult brain; some of the identified gene and pathway changes are relevant to older ages, including the altered SatB2 neuron location, delayed myelination and a role in NMDA receptor activation; Ash1L-KO causes ASD and ID phenotypes in the adult mice, all suggesting that at least some of the molecular changes may occur beyond the NPC stage. Validating some RNA seq changes in vivo would increase the significance of the data.

The authors state "we only examined the transcriptome changes 24 hours after induced differentiation, which might reflect the limited number of genes that had impaired expression in the Ash1L-KO cells during very early stage of NPC differentiation".

How does this relate to the changes that cause the ASD and ID like behaviors? What is the basis for this time selection?

Reviewer #3 (Remarks to the Author):

I appreciate the detailed response from the authors, and agree that the RT-PCR validation data are very convincing, especially at late time point. However, I think with such big difference, in vivo validation is worthy a try, even with the concern of multiple cell types. If any, in vitro induced differentiation usually result in mixed cell populations as well, especially astrocytes after long time culture.

Also, I am still not convinced to only look into up-regulated genes during differentiation because Ash1 is considered as a transcriptional activator. Simple analysis like Weighted Gene Co-expression network analysis (WGCNA) should be able to offer unbiased information about the effects of Ash1 KO on differentiation, especially with data from multiple time points.

Response to reviewers:

We would like to thank the reviewers' constructive comments for our manuscript, which largely improves the overall quality of this study. We have taken the reviewers' comments into careful consideration and added new data to address reviewers' comments in the revised manuscript. The followings are the point-by-point responses to reviewer's comments:

Reviewer #2 (Remarks to the Author):

1. It was recommended that heterozygotes be tested for ASD and ID phenotypes, rather than only homozygote *Ash1L* KOs, to increase the relevance to the human disorder. However, this was not included in the revision.

Response: We agree with the reviewer that heterozygous *Ash1L* disruptive mutation is more representative of situations of human patients. Considering the primary aim of current study was to prove the causality between *Ash1L* loss in the developing brain and genesis of ASD/ID, we did not include the heterozygous *Ash1L*-KO mice to quantitatively measure whether the degree of ASD/ID-like defects was correlated with *Ash1L* gene dosage in the developing brain. After discussing with editor, we agree this has been appropriately stated as one of limitations of current study in the Discussion section (page 9).

2. Validating some RNA seq changes in vivo.

Response: To validate the results *in vivo*, we performed the qRT-PCR analysis to examine the expression of five NDD-related genes in wild-type and *Ash1L*-KO E16.5 cortices. The results showed that *Emx2*, *Pcdh10*, and *Foxg1* had reduced expression in both *Ash1L*-KO cortices and differentiating NPCs (Fig. 4E-I), while *Dbx2* and *Sall3* did not show significant difference in expression between wild-type and *Ash1L*-KO cortices (Fig. S4C and S4D).

Reviewer #3 (Remarks to the Author):

1. Validating some RNA seq changes in vivo.

Response: To validate the results *in vivo*, we performed the qRT-PCR analysis to examine the expression of five NDD-related genes in wild-type and *Ash1L*-KO E16.5 cortices. The results showed that *Emx2*, *Pcdh10*, and *Foxg1* had reduced expression in both *Ash1L*-KO cortices and differentiating NPCs (Fig. 4E-I), while *Dbx2* and *Sall3* did not show significant difference in expression between wild-type and *Ash1L*-KO cortices (Fig. S4C and S4D).

2. Further discussion of downregulated genes.

Response: Following the reviewer's comments, we analyzed the 2,808 genes downregulated in the differentiating wild-type NPCs and identified 44 genes upregulated in the *Ash1L*-KO cells (cutoff: fold changes > 1.5, $p < 0.01$) (Fig. 4A), and the GO enrichment analysis showed that they

were involved in cell migration, motility, and developmental growth (cutoff: FDR < 0.05) (Fig. 4B, Table S3). We further discuss the up- and downregulated genes in the Discuss section (page 8).

All revised parts in the manuscripts are highlighted as **dark red**.